# Predictors of elevated carotid intima-media thickness as a surrogate marker for cardiovascular disease in pediatric chronic kidney disease

Emi Yulianti[1], Retno Palupi-Baroto[1], Sasmito Nugroho[1], Noormanto Noormanto[1], Maria Grace Wilianto[2], Dwi Astuti Dharma Putri[3], Indah Kartika Murni [1]*

1 Department of Child Health, Faculty of Medicine, Public Health and Nursing, Universitas Gadjah Mada/ Dr Sardjito Hospital, Yogyakarta, Indonesia, 2 Faculty of Medicine, Universitas Kristen Duta Wacana, Yogyakarta, Indonesia, 3 Faculty of Medicine, Public Health and Nursing, Universitas Gadjah Mada, Yogyakarta, Indonesia

* indah.kartika.m@ugm.ac.id

## Background

Chronic kidney disease (CKD), especially in its late stages, carries significant morbidity and mortality, often due to cardiovascular problems. Mortality rates for children with CKD and cardiovascular disorders remain high even in high-income countries. Elevated carotid intima-media thickness (CIMT) is considered a marker for vascular thickness and future cardiovascular events in younger populations. This study aimed to determine predictors of cardiovascular events in pediatric patients with CKD with CIMT as a surrogate marker.

## Materials and methods

A retrospective cohort study was conducted on children aged 2–18 years with CKD at Dr. Sardjito Hospital, Yogyakarta, Indonesia, from September 1st 2022–31st December 2023. We used multivariate logistic regression to identify independent predictors at diagnosis (e.g., age, male sex, obesity, CKD stage, and hypertension) associated with elevated CIMT (outcome), measured at least three months post-diagnosis as a proxy for cardiovascular risk.

## Results

A total of 71 patients were recruited, 35 (49.3%) of whom were male, and the median age (range) was 14.67 years (4.6–18.8). Twenty-four children (33.8%) had increased vascular thickness. In the multivariate analysis, male sex was independently associated with increased vascular thickness with an adjusted odds ratio (95% CI) of 2.911 (1.012–8.371, p = 0.047).

## Conclusion

Approximately one in three children with CKD experienced increased vascular thickness. Male was an independent predictor for increased vascular thickness in children with CKD.

**Data availability statement:** All relevant data are within the manuscript.

**Funding:** The author(s) received no specific funding for this work.

**Competing interests:** The authors have declared that no competing interests exist.

## Introduction

Chronic kidney disease (CKD), especially in its late stages, has been associated with significant morbidity and mortality, often due to cardiovascular problems. Complications due to cardiovascular events are considered quite significant in children with CKD because they possess a higher rate of mortality compared to the mortality caused by CKD itself [1,2]. Even in high-income countries like the United States and Australia where healthcare facilities are assumed to be more advanced, the mortality rates for children with CKD and cardiovascular disorders remain high, at 23% and 50% respectively [2].

The elevated mortality risk is largely attributed to factors such as accelerated atherosclerosis, vascular calcification, and chronic inflammation, which are common in CKD patients. Among these, cardiovascular disease stands out as a major in individuals with end-stage CKD. Both accelerated atherosclerosis and cardiovascular disease are major causes of morbidity and mortality in patients with end-stage CKD [3,4]. In 2018, mortality rates for children aged 1–19 years with end-stage CKD in general US pediatric population were reported at 0.31 per 1000 population. Among 1,454 pediatric dialysis patients, 31.2% developed cardiac-related events [5]. The most commonly reported cardiovascular events resulting in death in children with CKD were arrhythmia (19.6%), heart valvular disease (11.7%), cardiomyopathy (9.6%), and sudden cardiac death (2.8%) [2].

Atherosclerosis is one important stage of cardiovascular cascade events, significantly contributing to morbidity and mortality [2]. Subclinical atherosclerosis is an early indicator of atherosclerotic problems [6]. Carotid-intima media thickness (CIMT) measured by a Doppler ultrasound is established a safe, non-invasive, and cost-effective marker for detecting early-stage or subclinical atherosclerosis [7]. CIMT reflects arterial wall thickness, making it a reliable diagnostic tool [8,9]. Further, elevated CIMT is also considered as a predictor of future cardiovascular events in younger populations [10,11].

In a cohort study of children with CKD, it was observed that the index of CIMT was elevated by 74.5%, with a higher incidence of 90.9% in CKD stages I and II [12]. Studies indicates that CIMT measurements can predict cardiovascular events in children, including myocardial infarction, coronary heart disease, and stroke. As far as we aware, limited studies have explored predictor factors for cardiovascular events in children with CKD in Indonesia. Therefore, this study aimed to determine predictors of cardiovascular events in pediatric patients with CKD using CIMT as a surrogate marker.

## Materials and methods

### Study settings and population

A retrospective cohort study was conducted at Dr. Sardjito Hospital, a national referral hospital in Yogyakarta province, Indonesia. The study included children aged 2–18 years diagnosed with CKD for at least three months, regardless of etiology, from September 2022 to December 2023. The participants were recruited from the Nephrology Clinic as outpatients or hospitalized patients at Dr. Sardjito Hospital.

The sample size calculation was based on a 74.5% prevalence of elevated CIMT in children with CKD [5], considering five independent variables. Following the rule of thumb for factor analysis (at least 10 events per predictor), a minimum of 67 patients was required.

Patients with incomplete medical records and complex congenital heart disease were excluded from the study. All eligible participants underwent ultrasound examinations to assess heart function parameters and a follow-up measure for CIMT at the Cardiology Clinic, Dr. Sardjito Hospital. A baseline clinical and echocardiography data were collected at the time of diagnosis and CIMT measurements were performed at least three months post-diagnosis to assess cardiovascular events indicated by vascular thickness. Data on previous cardiovascular events, age at diagnosis, nutritional status, dialysis stage, sex, and hypertension status were extracted from electronic medical records for analysis.

## Study definition

Nutritional status was evaluated using height and weight measurements, applying the WHO Growth Reference 2007 definitions through WHO AnthroPlus software [13]. Height measurement in children who can stand were measured using stadiometer with the child stands barefoot with heels, buttocks, shoulders, and head touching the vertical board. For children who were not able to stand, recumbent length is measured using an infantometer, with the child lying flat, and the head and feet aligned to the measurement board. While for weight measurement, a calibrated digital scale were used. Measurements were taken with minimal clothing and no shoes to avoid interference with the reading. The nutritional status classifications used were: "normal" for weight-for-length between −2 standard deviations (SD) and +2 SD; "underweight" for scores below −2 SD; "severely wasted" for scores less than −3 SD; "overweight" for scores above 1 SD up to 2 SD above the mean; and "obese" for scores exceeding 2 SD above the mean. This approach enabled standardized assessment of nutritional health, facilitating identification of individuals at risk of malnutrition or obesity within the cohort.

In this study, CKD stages were classified based on glomerular filtration rates (GFR) according to the National Kidney Foundation-KDOQI [14] guidelines. CKD stages were further categorized into two groups: predialysis and dialysis. Predialysis included stage 1–4 CKD (GFR 15 up to ≥90 mL/min/1.73m$^2$) or stage 5 CKD (GFR < 15 mL/min/1.73m$^2$) managed conservatively, while dialysis stage referred to participants undergoing dialysis due to stage 5 CKD. Blood pressure measurements were averaged from three readings taken after a ten-minute rest period. Elevated blood pressure, including hypertension, was defined as per the Clinical Practice Guidelines for Screening and Management of High Blood Pressure in Children and Adolescents by the American Academy of Pediatrics [15]. For adolescents aged ≥13 years, elevated blood pressure was indicated by a systolic blood pressure of ≥120 mmHg, regardless of diastolic blood pressure. This standardized approach ensured consistent assessment of CKD stages and blood pressure across the study cohort, facilitating accurate evaluation of cardiovascular risk factors.

## CIMT and echocardiographic examination

All included participants underwent an ultrasound examination to determine the CIMT and heart function parameters. As a part of single-blind study design, the measurement was done by trained technicians who were blinded to the participant's clinical information. Heart function parameters were left ventricular systolic and diastolic function, as well as right ventricular systolic function. These parameters were categorized as normal or abnormal according to the classification by Park et al. [16]. CIMT served as a marker for early vascular impairment in the atherosclerosis process defined as the combined thickness of media and intima layers of the carotid artery. The CIMT measurements were conducted using a common carotid artery B-mode ultrasound with a Phillips HD11 XE ultrasound system, a fully digital imaging platform designed for high-definition vascular imaging. It was equipped with an L12-3 broadband linear array transducer, which operates at frequencies of 3–12 MHz and features a 38 mm aperture for precise assessments of vascular structures. This combination enables detailed imaging and ensures accurate evaluation of CIMT in pediatric patients [16]. It was taken on the posterior (far) wall of the left carotid artery during the end-diastolic phase. To derive mean CIMT, at least three measurements were

taken approximately 10 mm proximal to the bifurcation. Abnormal/elevated CIMT score was defined as a value ≥ 95th percentile for age, sex, and height [17]. A baseline echocardiogram to determine heart function parameters at diagnosis was extracted from all patients through medical records.

## Statistical Analysis

The data was analyzed with IBM SPSS version 26 (StataCorp LP, Texas). Normality of numerical data distribution was assessed with the Shapiro-Wilk test or Kolmogorov-Smirnov. Data are presented as means (± SD) for normally distributed continuous data, median and interquartile range (IQR) for skewed data, and frequencies or percentages for categorical data. A T-test compared numerical variables, while Chi-square tests analyzed dichotomous data.

The independent variable was the predictors of elevated CIMT, while the dependent variable (outcome) was CIMT. Univariate and multivariate analysis logistic regression identified predictors of early ascular impairement in CKD patients. Variables with a p-value <0.25 in the univariate anaysis were considered for multivariate analysis, and those with p < 0.05 in the multivariate analysis were independent predictors.

## Ethical approval

Ethics approval was obtained from the Research Ethics Committee, Faculty of Medicine, Public Health and Nursing, Universitas Gadjah Mada, Indonesia (KE/FK/0876/EC/2022) and (KE/FK/2037/EC/2023). Written individual informed consent was taken from parents or guardians of selected participants. Data used in this study is available upon request.

## Results

A total of 71 participants were included during the study period with 50.7% were male. Median age was 14.67 years (IQR: 4.6–18.8) and more than half of the children had a normal weight for length score. Twenty-four children (33.8%) had increased vascular thickness. Except or dialysis stage, no significant differences were found in participant characteristics between those with increased CIMT and those with normal CIMT measurement (Table 1).

Ventricular hypertrophy was found in 59% of children diagnosed with CKD with concentric type as the most common type reported (Table 2). Systolic and diastolic functions showed no statistical differences between groups with and without elevated CIMT. In the multivariate analysis, male sex was independently associated with increased vascular thickness with an adjusted odds ratio (95% CI) of 2.911 (1.012–8.371, p = 0.047) (Table 3).

## Discussion

This cohort study examines the potential for cardiovascular disorders among children diagnosed with CKD at Dr. Sardjito Hospital using CIMT as a surrogate marker. Increased CIMT can signify progressive changes in arterial wall thickness over time. Male sex was identified as independent predictors associated with increased vascular thickness. These findings highlight the importance of monitoring CIMT as a tool for assessing cardiovascular risk in pediatric CKD patients.

Male sex is often associated with higher testosterone levels which might play a role in vascular thickening. A study found that boys are 2.06 times more likely to develop increased CIMT compared to girls. Boys also tend to have an increased prevalence of hypertension, even before puberty, attributed to testosterones effects on vasoconstriction and kidney sodium reabsorption via angotensin-1 receptors [18,19]. Conversely, endogenous estrogen levels in girls are considered a protective factor against cardiovascular events, lowering their risk after puberty [18]. A previous study within obese adolescents in Indonesia revealed a similar trend where boys tend to have higher odds of elevated CIMT [20]. It was stated that boys with elevated CIMT showed some metabolic risk factors such as increased values of LDL cholesterol and it was associated with greater odds of elevated CIMT. CIMT as a surrogate marker is also highly correlated with higher morbidity and mortality associated with atherosclerosis cascade [3].

**Table 1. Characteristics of Research Subjects.**

| Characteristics | | Total Subject (n=71) | Increased CIMT (n=24) | Normal CIMT (n=47) | p |
|---|---|---|---|---|---|
| Gender, n (%) | | | | | |
| | Male | 35 (49.3) | 16 (66.7) | 19 (40.4) | 0.036*ˣ |
| | Female | 36 (50.7) | 8 (33.3) | 28 (59.6) | |
| Age, year, median (min-max) | | 14.67 (4.6-18.8) | 15.52 (5.2-17.4) | 14.37 (4.6-18.8) | 0.279# |
| Age, n (%) | | | | | |
| | <10 years | 11 (15.5) | 3 (12.5) | 8 (17.0) | 0.759$ |
| | ≥10 years | 60 (84.5) | 21 (87.5) | 39 (83.0) | |
| Etiology of CKD, n (%) | | | | | |
| | CAKUT | 10 (14) | 5 (20.8) | 5 (10.6) | 0.497 |
| | GN | 49 (69) | 15 (62.5) | 34 (72.3) | |
| | Others | 12 (17) | 4 (16.7) | 8 (17) | |
| Nutritional status, n (%) | | | | | |
| | Normal | 40 (56.3) | 16 (66.7) | 24 (51.1) | 0.261# |
| | Wasted | 11 (15.5) | 3 (12.5) | 8 (17.0) | |
| | Severely wasted | 5 (7.0) | 1 (4.2) | 4 (8.5) | |
| | Overweight | 6 (8.5) | 1 (4.2) | 5 (10.6) | |
| | Obesity | 9 (12.7) | 3 (12.5) | 6 (12.8) | |
| CKD Stage, n (%) | | | | | |
| | Predialysis | 41 (57.7) | 10 (41.7) | 31 (66) | 0.050* |
| | Dialysis | 30 (42.3) | 14 (58.3) | 16 (34) | |
| Hipertension, n (%) | | | | | |
| | Yes | 43 (60.6) | 16 (66.7) | 27 (57.4) | 0.452ˣ |
| | No | 28 (39.4) | 8 (33.3) | 20 (42.6) | |
| Hb, mean±SD, mg/dL | | 11.00±2.58 | 10,80±2.09 | 11,10±2.81 | 0.648^ |

*) significant p<0.05, x) Chi-square, $) Fisher exact test, ^) Independent T-test, #) Mann-Whitney Hb: Hemoglobin, CKD: Chronic Kidney Disease, CAKUT: Congenital Anomaly of the Kidney and Urinary Tract, GN: Glomerulonephritis, SD: Standard Deviation

Concentric-type ventricular hypertrophy was found as the majority of abnormality in heart parameters examination and is associated with hypertension. This aligns, with the findings of this study indicating that 60.6% of children with CKD had hypertension [6]. Poor management of blood pressure and CKD progression can lead to ongoing cardiac ventricular hypertrophy until functional impairment ensues. Notably, this study observed impaired heart function in only one participant. These findings underscore the critical importance of effectively managing blood pressure to mitigate cardiovascular complications in pediatric CKD patients.

A 2023 case-control study involving children with CKD stages 3–4 reported contradictory findings regarding the predictive value of CIMT. The study found no significant differences in CIMT measurements between CKD cases and healthy controls [21]. In contrast, a study in adults with advanced CKD, CIMT was significantly associated with lower estimated glomerular filtration rate (eGFR) and increased cardiovascular risk, supporting its role as a predictive marker [22]. However, in pediatric CKD patients, CIMT showed no significant correlation with CKD stage or cardiovascular risk factors [23]. Meanwhile, findings in predialysis CKD were mixed—some studies supported CIMT as a predictor of cardiovascular events, while others found carotid plaque burden to be more reliable. These inconsistencies highlight the need for population-specific approaches and further research to clarify CIMT's value in CKD management [24].

**Table 2. Echocardiography findings in children with CKD.**

| Echocardiography parameters | Total subject (n = 71) | Increased cIMT (n = 24) | normal cIMT (n = 47) | p |
|---|---|---|---|---|
| LV Mass, g, mean±SD | 126.96±6.07 | 143.16±73.15 | 118.68±66.16 | 0.159 |
| Increased LV Mass, n (%) | 48 (67.6) | 19 (79.2) | 29 (61.7) | 0.137 |
| LVMI, g/m², median (min-max) | 89.18 (45.6-361.1) | 88.95 (47.5-228) | 89.18 (45.6-361.1) | 0.277 |
| Increased RWT, n (%) | 35 (49.3) | 12 (50) | 23 (48.9) | 0.932 |
| Structural heart abnormalities, n (%) | | | | |
| Concentric left ventricular hypertrophy | 24 (33.8) | 11 (45.8) | 13 (27.6) | 0.205 |
| Eccentric left ventricular hypertrophy | 18 (25.3) | 5 (20.8) | 13 (27.6) | |
| Concentric left ventricular remodeling | 11 (15.5) | 1 (4.2) | 10 (21.3) | |
| Systolic dysfunction, n(%) | 5 (7.0) | 1 (4.2) | 4 (8.5) | 0.656 |
| FS, %, median (min-max) | 34.8 (16.4–55.2) | 34.75 (24–44.5) | 34.8 (16.4-55.2) | 0.795 |
| Abnormal FS, n (%) | 41 (57.7) | 14 (58.3) | 27 (57.4) | 0.943 |
| MV E/A, median (min-max) | 1.5 (0.8–2.3) | 1.5 (0.8–2) | 1.5 (0.9-2.3) | 0.920 |
| Diastolic dysfunction, n (%) | 2 (2.8) | 1 (4.2) | 1 (2.1) | 1.000 |
| TAPSE, mm, median (min-max) | 2.1 (1.4–3.0) | 2.1 (1.4–2.9) | 2.1 (1.4-3) | 0.664 |
| Abnormal TAPSE, n (%) | 50 (73.5) | 14 (66.7) | 36 (76.6) | 0.391 |

LV: left ventricle; LVMI: left ventricle mass index; EF: ejection fraction; FS: fractional shortening; MV E/A: mitral valve E/A; TAPSE: Tricuspid Annular Plane Systolic Excursion; RWT: relative wall thickness.

**Table 3. Predictors of cardiovascular disorders in children with CKD.**

| | | CIMT | | | | Bivariate Analysis | | Multivariate Analysis | |
|---|---|---|---|---|---|---|---|---|---|
| | | Yes | | No | | OR (CI 95%) | p | *Adjusted* OR (IK 95%) | p |
| | | n | % | n | % | | | | |
| Age | | | | | | | | | |
| | ≥10 years | 21 | 35.0 | 39 | 65.0 | 1.8 (0.46–3.57) | 0.739$ | | |
| | <10 years | 3 | 27.3 | 8 | 72.7 | | | | |
| Gender | | | | | | | | | |
| | Male | 16 | 45.7 | 19 | 54.3 | 2.06 (1.01–4.18) | 0.047 | 2.911 (1.012–8.371) | 0.047* |
| | Female | 8 | 22.2 | 28 | 77.8 | | | | |
| Obesity | | | | | | | | | |
| | Yes | 3 | 33.3 | 6 | 66.7 | 0.98 (0.37–2.64) | 1.000$ | | |
| | No | 21 | 33.9 | 41 | 66.1 | | | | |
| CKD Stage | | | | | | | | | |
| | Dialysis | 14 | 46.7 | 16 | 53.3 | 1.91 (0.99-3.71) | 0.064 | 0.374 (0.132–1.061) | 0.064 |
| | Predialysis | 10 | 24.4 | 31 | 75.6 | | | | |
| Hypertension | | | | | | | | | |
| | Yes | 16 | 37.2 | 27 | 62.8 | 1.30 (0.65–2.63) | 0.452ˣ | | |
| | No | 8 | 28.6 | 20 | 71.4 | | | | |

OR: Odd Ratio, *) statistically significant, x) Chi-square, $) Fisher exact test

To support practical clinical integration, CIMT assessment should become part of routine cardiovascular risk evaluation in children with CKD. Annual CIMT screening is recommended for those with CKD stages 3–5, and every two years for stages 1–2. CIMT values at or above the 95th percentile (adjusted for age, sex, and height) are considered abnormal and

signal increased cardiovascular risk. An elevated CIMT should trigger further evaluation, including blood pressure monitoring and echocardiographic assessment for cardiac structure and function. Children with both elevated CIMT and other risk factors, such as hypertension, should be prioritized for intensified interventions, including tighter blood pressure control, lifestyle changes, and targeted pharmacologic therapy. This integrated approach supports early identification and management of high-risk patients, aiming to improve long-term cardiovascular outcomes in pediatric CKD.

Patients with CKD and cardiovascular risk factors are encouraged to increase physical activity as part of their management. Simple activities such as 30 minutes of daily walking or monitoring daily steps with a goal of reaching 10,000 steps per day are beneficial. It's important to avoid exercise in cases of chest pain or a recent cardiac event within the last 6 weeks. Exercise should also be limited if there is resting shortness of breath, significant edema from fluid retention, acute infection or fever, diabetes, or uncontrolled blood pressure [25]. These guidelines help ensure safe physical activity tailored to the individual health status of CKD patients with cardiovascular concerns.

This study comprehensively examines cardiovascular risk across all stages of chronic kidney disease (CKD) in children. However, it has several limitations. Firstly, being conducted at Dr. Sardjito Hospital, a tertiary and national referral center, introduces potential recruitment bias since many patients are referred and may be in more advanced CKD stages compared to other settings. Our cohort compromised predominantly Southeast Asian children, which may limit generalizability to populations with differing CIMT reference norms. Future multiracial studies might be needed to validate a more universal CIMT thresholds. Additionally, CIMT measurements were only taken once, limiting the ability to track potential changes or further increases in CIMT thickness over time. Furthermore, the study did not account for other predictors such as dietary habits, physical activity levels, or genetic predisposition, which could influence vascular thickness independently of CKD. These limitations suggest the need for cautious interpretation of the findings regarding generalizability and longitudinal CIMT trends, and the role of external risk factors in pediatric CKD patients.

CIMT examination is valuable for assessing cardiovascular risk, particularly in identifying subclinical atherosclerosis and implementing preventive measures to reduce future cardiovascular events. This study recommends more frequent and comprehensive follow-up examinations for CKD children at high risk for cardiovascular issues in the future.

CIMT measurement serves as a predictive biomarker for diagnosing and prognosing cardiovascular disease in CKD children. Ultrasonography for CIMT assessment can detect atherosclerosis and assess coronary artery disease risk even in early CKD stages, demonstrating its precision and accuracy in disease diagnosis. B-mode ultrasound is a viable option due to its ease, affordability, and non-invasive nature, making CIMT measurement a practical addition to routine care for patients with CKD.

## Conclusion

One-third of pediatric patients with CKD showed thickening of carotid intima-media thickness (CIMT). Male gender was identified as significant predictors of cardiovascular disorders among children with CKD. These findings highlight the importance of monitoring CIMT and considering gender-specific in assessing cardiovascular risk in pediatric CKD patients.

## Acknowledgments

The authors would like to express their gratitude to Brigitta Beata for collecting data. The authors would also like to thank Adiel Christian Saputra, Muhammad Jalul Mutaqorrib, Zulfikar Ihyauddin, and Gita Novalia for editing.

## Author contributions

**Conceptualization:** Emi Yulianti, Retno Palupi-Baroto, Sasmito Nugroho, Noormanto Noormanto, Indah Murni.

**Data curation:** Emi Yulianti, Maria Grace Wilianto.

**Formal analysis:** Emi Yulianti, Retno Palupi-Baroto, Dwi Astuti Dharma Putri.

**Investigation:** Emi Yulianti, Maria Grace Wilianto.

**Methodology:** Emi Yulianti, Retno Palupi-Baroto, Sasmito Nugroho, Noormanto Noormanto, Indah Murni.

**Project administration:** Maria Grace Wilianto.

**Resources:** Emi Yulianti.

**Supervision:** Emi Yulianti, Retno Palupi-Baroto, Sasmito Nugroho, Noormanto Noormanto, Indah Murni.

**Validation:** Emi Yulianti, Retno Palupi-Baroto, Maria Grace Wilianto.

**Visualization:** Emi Yulianti, Maria Grace Wilianto, Dwi Astuti Dharma Putri.

**Writing – original draft:** Emi Yulianti, Indah Murni.

**Writing – review & editing:** Emi Yulianti, Retno Palupi-Baroto, Sasmito Nugroho, Noormanto Noormanto, Maria Grace Wilianto, Dwi Astuti Dharma Putri, Indah Murni.

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
