## [Decision Letter · Decision Letter 0]

PONE-D-24-25206Predictors of increased vascular thickness in children with chronic kidney diseasePLOS ONE

Dear Dr. Murni,

Thank you for submitting your manuscript to PLOS ONE. After careful consideration, we feel that it has merit but does not fully meet PLOS ONE’s publication criteria as it currently stands. Therefore, we invite you to submit a revised version of the manuscript that addresses the points raised during the review process.

We look forward to receiving your revised manuscript.

Kind regards,

Elena Olmastroni

Academic Editor

PLOS ONE

2. Please amend either the abstract on the online submission form (via Edit Submission) or the abstract in the manuscript so that they are identical.

Reviewers' comments:

Reviewer's Responses to Questions

**Comments to the Author**

1. Is the manuscript technically sound, and do the data support the conclusions?

Reviewer #1: Partly

Reviewer #2: Partly

2. Has the statistical analysis been performed appropriately and rigorously? 

Reviewer #1: Yes

Reviewer #2: No

3. Have the authors made all data underlying the findings in their manuscript fully available?

Reviewer #1: No

Reviewer #2: Yes

4. Is the manuscript presented in an intelligible fashion and written in standard English?

Reviewer #1: Yes

Reviewer #2: Yes

5. Review Comments to the Author

Reviewer #1: Thank you very much for inviting me as a peer reviewer and giving me the opportunity to review. The author studied the predictive factors of increased vascular thickness in children with chronic kidney disease. This research is of great significance for reducing cardiovascular events in children with CKD in the future. The content of the article still needs further modification to arouse more interest from readers.

Introduction

There is a lack of connection between the first paragraph and the second paragraph. The first paragraph of this article tells the reason for the high mortality rate of CKD. The second paragraph directly leads to the acceleration of atherosclerosis and cardiovascular disease. The lack of connection of relevant content may confuse readers. Suggest the author to add reasons (such as high CKD mortality rate, which may be caused by factors such as XX, XX, etc., with cardiovascular factors being the main factor).

Methods

1. The sample size of this study is 71 cases. How did the author determine the sample size? It is not provided in the manuscript.

2. Was blinding used in this study? If used, is it single blind or double-blind? Please explain to the author.

3.What is the deadline for measuring CIMT at least three months after CKD diagnosis? Is follow-up conducted after CIMT and echocardiography measurements? If so, how long is the follow-up period?

Results

Could the author explain the main outcome measure of this study? Is it CIMT? Please explain and make corresponding modifications to the abstract section.

Discussion

What is the possible reason for the statistical difference in the systolic and diastolic function of the heart between the CIMT elevated group and the non elevated group? Please add relevant literature to enhance the persuasiveness of the content.

Other parts

Is the relevant data in this article shared? Please indicate at the end of the article that the author.

Reviewer #2: Thank you for inviting me to review this manuscript. Overall, the study gets 2 out of 5 points regarding its novelty. I think the manuscript needs a major revision and cannot be considered for publication without addressing the following points:

-The title of the study does not accurately reflect its content and should be revised for better alignment.

- remove the sentences saying that this study is the first study on this topic.

-The authors describe their study as a retrospective cohort; however, the duration of follow-up is unclear. If no follow-up was conducted, please clarify how the relative risk (RR) was calculated.

-Line 46: The time period for which this mortality rate applies is missing. Please specify the year.

-Methods: It is recommended to include details about the ultrasound device used in the study.

-Methods: A reference for the assessment of nutritional status should be cited.

-Methods: Please provide details on how height and weight measurements were conducted.

-The limitations section should be expanded, as the study has additional limitations that were not fully addressed.

6. PLOS authors have the option to publish the peer review history of their article (what does this mean? ). If published, this will include your full peer review and any attached files.

**Do you want your identity to be public for this peer review?** For information about this choice, including consent withdrawal, please see our Privacy Policy .

Reviewer #1: No

Reviewer #2: No

---

## [Author Response · Author response to Decision Letter 1]

16 Apr 2025

PONE-D-24-25206

Predictors of increased vascular thickness in children with chronic kidney disease

Response to the reviewers

Reviewer 1

Comments to the Author

Thank you very much for inviting me as a peer reviewer and giving me the opportunity to review. The author studied the predictive factors of increased vascular thickness in children with chronic kidney disease. This research is of great significance for reducing cardiovascular events in children with CKD in the future. The content of the article still needs further modification to arouse more interest from readers.

Introduction?

There is a lack of connection between the first paragraph and the second paragraph. The first paragraph of this article tells the reason for the high mortality rate of CKD. The second paragraph directly leads to the acceleration of atherosclerosis and cardiovascular disease. The lack of connection of relevant content may confuse readers. Suggest the author to add reasons (such as high CKD mortality rate, which may be caused by factors such as XX, XX, etc., with cardiovascular factors being the main factor).

Authors’ response

Thank you for the suggestion to revise the wording and flow of the manuscript. We have amended the sentences as suggested.

Change made in the manuscript

The elevated mortality risk is largely attributed to factors such as accelerated atherosclerosis, vascular calcification, and chronic inflammation, which are common in CKD patients. Among these, cardiovascular disease stands out as a major in individuals with end-stage CKD. (Page 3, line 45 – 47)

Methods?

1. The sample size of this study is 71 cases. How did the author determine the sample size? It is not provided in the manuscript.

2. Was blinding used in this study? If used, is it single blind or double-blind? Please explain to the author.

3. What is the deadline for measuring CIMT at least three months after CKD diagnosis? Is follow-up conducted after CIMT and echocardiography measurements? If so, how long is the follow-up period?

Authors’ response

Thank you for the suggestion to improve the method section. We have modified the related points as suggested.

1. Sample size calculation have been added in the method section.

2. This study is a single blind study who is the trained technicians conducting the echocardiography assessment, we added this information within the manuscript.

3. No follow up period after CIMT measurement was conducted within this study, this limitation was listed within the discussion.

Change made in the manuscript

1. The sample size calculation was based on a 74.5% prevalence of elevated CIMT in children with CKD, considering five independent variables. Following the rule of thumb for factor analysis (at least 10 events per predictor), a minimum of 67 patients was required (Page 4, line 78 – 80)

2. As a part of single-blind study design, the measurement was done by trained technicians who were blinded to the participant’s clinical information. (Page 6, line 120-121)

3. None

Results?

Could the author explain the main outcome measure of this study? Is it CIMT? Please explain and make corresponding modifications to the abstract section.

Authors’ response

This study aimed to identify predictors of cardiovascular events in pediatric patients with CKD, using CIMT as a surrogate marker. The independent variable was the predictors of elevated CIMT, while the dependent variable (outcome) was CIMT. This outcome has been incorporated into the abstract (last sentence of the background), as well as the introduction and methods sections.

Change made in the manuscript

Abstract:

Carotid intima-media thickness (CIMT) was the outcome measure, assessed at least three months post-diagnosis to evaluate cardiovascular events based on vascular thickness (Page 2, line 24-26).

Method section:

The independent variable was the predictors of elevated CIMT, while the dependent variable (outcome) was CIMT (Page 7, line 144-145).

Discussion?

What is the possible reason for the statistical difference in the systolic and diastolic function of the heart between the CIMT elevated group and the non elevated group? Please add relevant literature to enhance the persuasiveness of the content.

Authors’ response

As provided in the Result section, page 10 line 169-170, this study revealed that systolic and diastolic functions showed no statistical differences between groups with and without elevated CIMT.

Change made in the manuscript

None

Other parts?

Is the relevant data in this article shared? Please indicate at the end of the article that the author.

Authors’ response

Data used in this study is available upon request.

Change made in the manuscript

Data used in this study is available upon request. (Page 8, line 154)

Reviewer 2

Comments to the Author

Thank you for inviting me to review this manuscript. Overall, the study gets 2 out of 5 points regarding its novelty. I think the manuscript needs a major revision and cannot be considered for publication without addressing the following points:

1. The title of the study does not accurately reflect its content and should be revised for better alignment.

Authors’ response

Thank you for the feedback, we have modified the title and replaced it with “Predictors of elevated carotid intima-media thickness as a surrogate marker for cardiovascular disease in pediatric chronic kidney disease”

Change made in the manuscript

Predictors of elevated carotid intima-media thickness as a surrogate marker for cardiovascular disease in pediatric chronic kidney disease (Page 1, Line 1-2)

2. Remove the sentences saying that this study is the first study on this topic.

Authors’ response

Thank you for the suggestion to remove the specific sentence, we have modified it as suggested.

Change made in the manuscript

As far as we are aware, limited studies have explored predictive factors for cardiovascular events in children with CKD in Indonesia (Page 4, Line 66-67).

3. The authors describe their study as a retrospective cohort; however, the duration of follow-up is unclear. If no follow-up was conducted, please clarify how the relative risk (RR) was calculated.

Authors’ response

Our retrospective cohort study used multivariate logistic regression to identify independent predictors at diagnosis (e.g., age, male sex, obesity, CKD stage, and hypertension) associated with elevated CIMT (outcome), measured at least three months post-diagnosis as a proxy for cardiovascular risk. Adjusted odds ratios (OR) were obtained from the analysis, as stated in the abstract. We have also corrected the RR in Table 3 as suggested.

Change made in the manuscript

Abstract: We used multivariate logistic regression to identify independent predictors at diagnosis (e.g., age, male sex, obesity, CKD stage, and hypertension) associated with elevated CIMT (outcome), measured at least three months post-diagnosis as a proxy for cardiovascular risk (Page 2, line 24-26).

Table 3: OR: Odd Ratio (Page 11, Line 182-183).

4. Line 46: The time period for which this mortality rate applies is missing. Please specify the year.

Authors’ response

Thank you for the kind correction, we have added the time period of the mortality rate in the manuscript.

Change made in the manuscript

In 2018, mortality rates for children aged 1-19 years with end-stage CKD in general US pediatric population were reported at 0.31 per 1000 population. (Page 3, Line 49-50)

5. Methods: It is recommended to include details about the ultrasound device used in the study.

Authors’ response

Thank you for the recommendation, we have added the details of the ultrasound device in the method section.

Change made in the manuscript

The CIMT measurements were conducted using a common carotid artery B-mode ultrasound with a Phillips HD11 XE ultrasound system, a fully digital imaging platform designed for high-definition vascular imaging. It was equipped with an L12-3 broadband linear array transducer, which operates at frequencies of 3–12 MHz and features a 38 mm aperture for precise assessments of vascular structures. This combination enables detailed imaging and ensures accurate evaluation of CIMT in pediatric patients [14]. (Page 6-7, Line 125-131)

6. Methods: A reference for the assessment of nutritional status should be cited.

Authors’ response

We have added the citation for nutritional status assessment in the method section as advised.

Change made in the manuscript

Nutritional status was evaluated using height and weight measurements, applying the WHO Growth Reference 2007 definitions through WHO AnthroPlus software [11] (Page 5, Line 91-92).

7. Methods: Please provide details on how height and weight measurements were conducted.

Authors’ response

Thank you for the advise, we have added the details of related measurements as suggested.

Change made in the manuscript

Height measurement in children who can stand were measured using stadiometer with the child stands barefoot with heels, buttocks, shoulders, and head touching the vertical board. For children who were not able to stand, recumbent length is measured using an infantometer, with the child lying flat, and the head and feet aligned to the measurement board. While for weight measurement, a calibrated digital scale were used. Measurements were taken with minimal clothing and no shoes to avoid interference with the reading. The nutritional status classifications used were… (Page 5, Line 92-98).

8. The limitations section should be expanded, as the study has additional limitations that were not fully addressed.

Authors’ response

Thank you for your review, we have expanded the limitations in the discussion section.

Change made in the manuscript

Furthermore, the study did not account for other predictors such as dietary habits, physical activity levels, or genetic predisposition, which could influence vascular thickness independently of CKD. These limitations suggest the need for cautious interpretation of the findings regarding generalizability and longitudinal CIMT trends, and the role of external risk factors in pediatric CKD patients. (Page 13, Line 225-229)

---

## [Decision Letter · Decision Letter 1]

PONE-D-24-25206R1Predictors of elevated carotid intima-media thickness as a surrogate marker for cardiovascular disease in pediatric chronic kidney diseasePLOS ONE

Dear Dr. Murni,

Thank you for submitting your manuscript to PLOS ONE. After careful consideration, we feel that it has merit but does not fully meet PLOS ONE’s publication criteria as it currently stands. Therefore, we invite you to submit a revised version of the manuscript that addresses the points raised during the review process.

We look forward to receiving your revised manuscript.

Kind regards,

Elena Olmastroni

Academic Editor

PLOS ONE

Journal Requirements:

Reviewers' comments:

Reviewer's Responses to Questions

**Comments to the Author**

1. If the authors have adequately addressed your comments raised in a previous round of review and you feel that this manuscript is now acceptable for publication, you may indicate that here to bypass the “Comments to the Author” section, enter your conflict of interest statement in the “Confidential to Editor” section, and submit your "Accept" recommendation.

Reviewer #1: All comments have been addressed

Reviewer #2: (No Response)

2. Is the manuscript technically sound, and do the data support the conclusions?

Reviewer #1: Yes

Reviewer #2: (No Response)

3. Has the statistical analysis been performed appropriately and rigorously? 

Reviewer #1: Yes

Reviewer #2: (No Response)

4. Have the authors made all data underlying the findings in their manuscript fully available?

Reviewer #1: Yes

Reviewer #2: (No Response)

5. Is the manuscript presented in an intelligible fashion and written in standard English?

Reviewer #1: Yes

Reviewer #2: (No Response)

6. Review Comments to the Author

Reviewer #1: Thank you to the author for responding carefully to the first round of review comments and completing the revisions. The article has made significant improvements in methodological description, data presentation, and discussion depth, but there are still the following issues that need further improvement:

Discussion section of the article:

1. Need to respond to the latest evidence-based medicine evidence: contradictory conclusions regarding the predictive value of CIMT compared to CKD studies published in 2023

2. Clinical translation suggestion: Provide specific pathways for integrating CIMT testing into the existing cardiovascular risk assessment system for CKD children, such as screening frequency and threshold definition.

3.Given the racial differences in the normal reference values of CIMT in children, it is recommended to supplement the racial composition of the study population in the limitations section.

Reference section:

Due to some of the references being relatively old, it is recommended that the author update 3-5 relevant references from the past 3 years.

Reviewer #2: (No Response)

7. PLOS authors have the option to publish the peer review history of their article (what does this mean? ). If published, this will include your full peer review and any attached files.

**Do you want your identity to be public for this peer review?** For information about this choice, including consent withdrawal, please see our Privacy Policy .

Reviewer #1: No

Reviewer #2: No

---

## [Author Response · Author response to Decision Letter 2]

18 Jun 2025

PONE-D-24-25206R1

Predictors of elevated carotid intima-media thickness as a surrogate marker for cardiovascular disease in pediatric chronic kidney disease

Response to the reviewers

Reviewer #1

Comments to the Author

Thank you to the author for responding carefully to the first round of review comments and completing the revisions. The article has made significant improvements in methodological description, data presentation, and discussion depth, but there are still the following issues that need further improvement:

Discussion section of the article:

1. Need to respond to the latest evidence-based medicine evidence: contradictory conclusions regarding the predictive value of CIMT compared to CKD studies published in 2023

Authors’ response

Thank you for the kind suggestion to respond to the latest evidence-based medicine evidence. We have added and cited the article as suggested into the discussion section.

Change made in the manuscript

A 2023 case-control study involving children with CKD stages 3-4 reported contradictory findings regarding the predictive value of CIMT. The study found no significant differences in CIMT measurements between CKD cases and healthy controls [21]. In contrast, a study in adults with advanced CKD, CIMT was significantly associated with lower estimated glomerular filtration rate (eGFR) and increased cardiovascular risk, supporting its role as a predictive marker [22]. However, in pediatric CKD patients, CIMT showed no significant correlation with CKD stage or cardiovascular risk factors [23]. Meanwhile, findings in predialysis CKD were mixed—some studies supported CIMT as a predictor of cardiovascular events, while others found carotid plaque burden to be more reliable. These inconsistencies highlight the need for population-specific approaches and further research to clarify CIMT’s value in CKD management [24]. (Page 13, line 212 – 221)

Discussion section of the article:

2. Clinical translation suggestion: Provide specific pathways for integrating CIMT testing into the existing cardiovascular risk assessment system for CKD children, such as screening frequency and threshold definition.

Authors’ response

Thank you for the suggestion to enrich the discussion in this article. As recommended, we have added to the manuscript a specific pathway that integrates CIMT examination into the current system for assessing cardiovascular in children with CKD.

Change made in the manuscript

To support practical clinical integration, CIMT assessment should become part of routine cardiovascular risk evaluation in children with CKD. Annual CIMT screening is recommended for those with CKD stages 3-5, and every two years for stages 1-2. CIMT values at or above the 95th percentile (adjusted for age, sex, and height) are considered abnormal and signal increased cardiovascular risk. An elevated CIMT should trigger further evaluation, including blood pressure monitoring and echocardiographic assessment for cardiac structure and function. Children with both elevated CIMT and other risk factors, such as hypertension, should be prioritized for intensified interventions, including tighter blood pressure control, lifestyle changes, and targeted pharmacologic therapy. This integrated approach supports early identification and management of high-risk patients, aiming to improve long-term cardiovascular outcomes in pediatric CKD. (Page 13, line 223 – 232)

Discussion section of the article:

3. Given the racial differences in the normal reference values of CIMT in children, it is recommended to supplement the racial composition of the study population in the limitations section.

Authors’ response

Thank you for highlighting the importance of considering racial differences in measuring CIMT in children. As suggested, we have included this issue on the limitations section.

Change made in the manuscript

Our cohort compromised predominantly Southeast Asian children, which may limit generalizability to populations with differing CIMT reference norms. Future multiracial studies might be needed to validate a more universal CIMT thresholds. (Page 14, line 246 – 249)

Reference section:

Due to some of the references being relatively old, it is recommended that the author update 3-5 relevant references from the past 3 years.

Authors’ response

Thank you for the suggestion. We have added updated relevant references from the past three years to statements previously supported by relatively older articles. Additional new references have also been included to support Discussion section, as outlined in response #1 to reviewer.

Change made in the manuscript

Within Introduction section:

Chronic kidney disease (CKD), especially in its late stages, has been associated with significant morbidity and mortality, often due to cardiovascular problems. Complications due to cardiovascular events are considered quite significant in children with CKD because they possess a higher rate of mortality compared to the mortality caused by CKD itself [1,2]. (Page 3, line 40) � previously only supported by reference number one from Shulman et al. (1989)

The elevated mortality risk is largely attributed to factors such as accelerated atherosclerosis, vascular calcification, and chronic inflammation, which are common in CKD patients. Among these, cardiovascular disease stands out as a major in individuals with end-stage CKD. Both accelerated atherosclerosis and cardiovascular disease are major causes of morbidity and mortality in patients with end-stage CKD [3,4]. (Page 3, line 49) � previously only supported by reference number three from Kumar et al. (2009)

CIMT reflects arterial wall thickness, making it a reliable diagnostic tool [8,9]. (Page 4, line 60) � previously only supported by reference number eight from Chen et al. (1999)

New added articles within the references list:

2. Li L-C, Tain Y-L, Kuo H-C, Hsu C-N. Cardiovascular diseases morbidity and mortality among children, adolescents and young adults with dialysis therapy. Front Public Health. 2023;11: 1142414. doi:10.3389/fpubh.2023.1142414

4. Marx-Schütt K, Cherney DZI, Jankowski J, Matsushita K, Nardone M, Marx N. Cardiovascular disease in chronic kidney disease. European Heart Journal. 2025; ehaf167. doi:10.1093/eurheartj/ehaf167

9. Akinmolayan A, Papacosta AO, Lennon LT, Ellins EA, Halcox JPJ, Whincup PH, et al. Carotid Intima‐Media Thickness, Carotid Distensibility, and Incident Heart Failure in Older Men: The British Regional Heart Study. JAHA. 2025; e037167. doi:10.1161/JAHA.124.037167

21. Hussain Z, Bajeer I, Khatri S, Mohsin S, Kumar P, Hashmi S. Carotid Intima-Media Thickness in Children with Chronic Kidney Disease as an Early Predictor of Cardiovascular Diseases. PJMHS. 2023;17: 65–68. doi:10.53350/pjmhs02023171265

22. Rizikalo A, Coric S, Matetic A, Vasilj M, Tocilj Z, Bozic J. Association of Glomerular Filtration Rate and Carotid Intima-Media Thickness in Non-Diabetic Chronic Kidney Disease Patients over a 4-Year Follow-Up. Life. 2021;11: 204. doi:10.3390/life11030204

23. Palupi-Baroto R, Hermawan K, Murni IK, Nurlita T, Prihastuti Y, Puspitawati I, et al. Carotid intima-media thickness, fibroblast growth factor 23, and mineral bone disorder in children with chronic kidney disease. BMC Nephrol. 2024;25: 369. doi:10.1186/s12882-024-03771-z

24. Colbert G, Jain N, De Lemos JA, Hedayati SS. Utility of Traditional Circulating and Imaging-Based Cardiac Biomarkers in Patients with Predialysis CKD. Clinical Journal of the American Society of Nephrology. 2015;10: 515–529. doi:10.2215/CJN.03600414

(Page 17 – 18, line 281 – 361)

---

## [Decision Letter · Decision Letter 2]

Predictors of elevated carotid intima-media thickness as a surrogate marker for cardiovascular disease in pediatric chronic kidney disease

PONE-D-24-25206R2

Dear Dr. Murni,

We’re pleased to inform you that your manuscript has been judged scientifically suitable for publication and will be formally accepted for publication once it meets all outstanding technical requirements.

Kind regards,

Elena Olmastroni

Academic Editor

PLOS ONE

Additional Editor Comments (optional):

Reviewers' comments:

Reviewer's Responses to Questions

**Comments to the Author**

1. If the authors have adequately addressed your comments raised in a previous round of review and you feel that this manuscript is now acceptable for publication, you may indicate that here to bypass the “Comments to the Author” section, enter your conflict of interest statement in the “Confidential to Editor” section, and submit your "Accept" recommendation.

Reviewer #1: All comments have been addressed

2. Is the manuscript technically sound, and do the data support the conclusions?

Reviewer #1: Yes

3. Has the statistical analysis been performed appropriately and rigorously? 

Reviewer #1: Yes

4. Have the authors made all data underlying the findings in their manuscript fully available?

Reviewer #1: Yes

5. Is the manuscript presented in an intelligible fashion and written in standard English?

Reviewer #1: Yes

6. Review Comments to the Author

Reviewer #1: (No Response)

7. PLOS authors have the option to publish the peer review history of their article (what does this mean? ). If published, this will include your full peer review and any attached files.

**Do you want your identity to be public for this peer review?** For information about this choice, including consent withdrawal, please see our Privacy Policy .

Reviewer #1: No

---

## [Editor Report · Acceptance letter]

PONE-D-24-25206R2

PLOS ONE

Dear Dr. Murni,

I'm pleased to inform you that your manuscript has been deemed suitable for publication in PLOS ONE. Congratulations! Your manuscript is now being handed over to our production team.

Kind regards,

on behalf of

Dr. Elena Olmastroni

Academic Editor

PLOS ONE